# Risk factors of progression to endometrial cancer in women with endometrial hyperplasia: A retrospective cohort study

Jin Young Jeong[1‡], Sung Ook Hwang[2‡], Banghyun Lee[2]*, Kidong Kim[3], Yong Beom Kim[3], Sung Hye Park[4], Hwa Yeon Choi[2]

**1** Hallym Research Institute of Clinical Epidemiology, Chuncheon-si, Gangwon-do, Republic of Korea, **2** Department of Obstetrics and Gynecology, Inha University Hospital, Inha University School of Medicine, Incheon, Republic of Korea, **3** Department of Obstetrics and Gynecology, Seoul National University Bundang Hospital, Seongnam-Si, Gyeonggi-Do, Republic of Korea, **4** Department of Obstetrics and Gynecology, Hallym University Kangdong Sacred Heart Hospital, Seoul, Republic of Korea

‡ Jin Young Jeong and Hwang Sung Ook contributed equally to this work and are co-first authors.
* banghyun.lee@gmail.com

## Abstract

### Objective

This study aimed to investigate risk factors of progression to endometrial cancer (EC) in women with non-atypical and atypical endometrial hyperplasia (EH).

### Methods

The data of 62,333 women with EH diagnostic codes from 2007 to 2018 were sourced from the Korean Health Insurance Review and Assessment Service databases. The data from 11,525 women with non-atypical EH and 2,219 women with atypical EH who met the selection criteria were extracted for analysis.

### Results

Risk of EC in women with EH decreased in 40–49 year olds compared to other ages (non-atypical EH: [≤39 vs. 40–49 years] HR, 0.557; 95% CI, 0.439–0.708; $P<0.001$; [≤39 vs. ≥50 years] $P = 0.739$; atypical EH: [≤39 vs. 40–49 years] HR, 0.391; 95% CI, 0.229–0.670; $P = 0.001$; [≤39 vs. ≥50 years] $P = 0.712$). Risk of EC increased with increase in number of follow-up biopsies in women with non-atypical EH (1 biopsy: HR, 1.835; 95% CI, 1.282–2.629; $P = 0.001$; ≥2 biopsies: HR, 3.644; 95% CI, 2.585–5.317; $P<0.001$) and in women receiving ≥2 follow-up biopsies with atypical EH (HR, 3.827; 95% CI, 1.924–7.612; $P = 0.001$). Time of progression to EC decreased in women ≥50 years old with non-atypical EH compared to other ages ($P = 0.004$) and showed no differences among ages in women with atypical EH ($P = 0.576$). Progestational agents were a protective factor for EC in women with non-atypical EH (HR, 0.703; 95% CI, 0.565–0.876; $P = 0.002$).

**Data Availability Statement:** The data used in this study were accessed from Health Insurance Review and Assessment Service (HIRA),

consisting of all data with diagnostic codes for endometrial hyperplasia (N85.0 and N85.1) generated between January 1, 2007 and February 28, 2018. Interested researchers can apply to gain access to the data at https://opendata.hira.or.kr/home.do.

**Funding:** This work was supported by INHA UNIVERSITY HOSPITAL Research Grant. The funders had no role in study design, data collection and analysis, decision to publish, or preparation of the manuscript.

**Competing interests:** The authors have declared that no competing interests exist.

## Conclusions

In this claim data analysis, women ≤39 and ≥50 years old with EH were at a high risk for progression to EC, and repeat follow-up biopsy after a diagnosis of EH increased detection of EC. Progestational agents were an effective modality to prevent EC in women with non-atypical EH.

## Introduction

Endometrial hyperplasia (EH) is a pathological condition characterized by proliferation of endometrial glandular and stromal structures. The revised World Health Organization (WHO) classification divides EH into non-atypical EH and atypical EH/endometrioid intrae-pithelial neoplasia without the previous simple and complex subtypes [1, 2]. EH, particularly with atypia, is a precursor to endometrial carcinoma (EC) [3]. In a study in the United States (US), the peak incidence of EH was 142/100,000 and 213/100,000 woman-years in simple and complex non-atypical EH, respectively (in subjects in their early 50s) and 56/100,000 woman-years in atypical EH (in subjects in their early 60s) [4].

Endometrial cancer (EC) is the most common cancer of the female reproductive tract [5]. The incidence of endometrial cancer has increased globally due to the increasing number of elderly people and increasing rates of obesity [6, 7].

In a retrospective study in which 170 women with EH were followed for a mean of 13.4 years (from 1 to 26.7 years), progression to EC occurred in 1.6% and 23% of women with non-atypical and atypical EH, respectively [2]. In a matched case-control study, cumulative risk of progression to EC at years 4, 9, and 19 after EH diagnosis was 1.2%, 1.9%, and 4.6%, respec-tively, in women with non-atypical EH and 8.2%, 12.4%, and 27.5% in women with atypical EH [8].

Several studies have reported that 10–59% of women with atypical EH had occult EC detected at hysterectomy [9]. Advanced age, menopause, obesity, diabetes mellitus, abnormal uterine bleeding, and (complex) atypical EH have been reported as predictive factors of con-current EC in women with EH [9–11]. However, clinical risk factors related to progression to EC in women with non-atypical and atypical EH have not been reported. Some studies have reported that serum DNA integrity index and molecular markers and immune cells related to the immune escape mechanisms may play roles in EC [12, 13]. Evaluating these roles in EH might be of benefit to predicting progression to EC in women with EH.

Management for EH has aimed to control symptoms such as heavy bleeding, detect concur-rent EC, and prevent subsequent development of EC [14]. However, no studies have evaluated risk factors that predict progression to EC in women with EH. Therefore, this study aimed to investigate factors that influence EH progression to EC.

## Materials and methods

### Study population and design

South Korea has a universal health coverage system, the National Health Insurance, that covers approximately 98% of the overall Korean population. The claims data of the Health Insurance Review and Assessment Service (HIRA) represent 46 million patients per year [15]. This claims data study used the HIRA databases and data with EH diagnostic codes generated between January 1, 2007 and February 28, 2018.

Inclusion criteria were women who had diagnostic codes for EH with procedure codes for endometrial biopsy within 60 days before or after an initial diagnostic code and women receiving 1 or more management approaches for EH at least 90 days after diagnosis of EH. Women who were diagnosed with EC or who underwent hysterectomy within 1 year after diagnosis of EH were excluded. Because the HIRA dataset uses anonymous identification codes to protect patients' personal information, approval of this study was waived by the Institutional Review Board of Inha University Hospital (No. 2019-11-015) on November 25, 2019.

## Data collection

The diagnostic codes in the 10th revision of the International Statistical Classification of Diseases and Related Health Problems (ICD-10) were used to obtain data for women who had been diagnosed with EH (N85.0: endometrial glandular hyperplasia and N85.1: endometrial adenomatous hyperplasia [atypia]). Simple or complex EH was not distinguished. The procedure codes were derived from health insurance medical care expense claim forms. The procedure codes for endometrial biopsy were dilatation and curettage (R4521), endometrial biopsy (C8571, C8572), aspiration biopsy (C8573), simple curettage (C8574), and hysteroscopic curettage (C8575). The procedure codes for hysterectomy were simple abdominal hysterectomy (R4143, R4147), complex abdominal hysterectomy (R4144, R4148), laparoscopic hysterectomy (simple [R0141] and complex [R0142]), subtotal hysterectomy (R4130), vaginal hysterectomy (simple [R4149] and complex [R4140]), and radical hysterectomy (R4154, R4155). EC was assigned in women who had related diagnostic codes (ICD-10: C54, C54.0, C54.1, C54.2, C54.3, C54.8, C54.9, C55) with procedure codes for endometrial biopsy or hysterectomy within 60 days before or after an initial diagnostic code or women who had diagnostic codes for EC more than 2 times within 1 year. In Korea, every person with a cancer diagnosis is registered with a unique code (called a C code) in the National Cancer Registry. This C code is used in all subsequent medical records and claims created for that patient. Therefore, cancer diagnosis based on claims is considered reliable [16]. Type 2 diabetes was defined as the presence of identical E11-E14 codes (ICD-10) at least twice for 1 patient or a diabetes drug code (including biguanides, sulfonylurea, meglitinides, thiazolidinediones, dipeptidyl peptidase-4 inhibitors, α-glucosidase inhibitors, sodium-glucose co-transporter 2 inhibitors, insulin, or glucagon-like peptide 1 agonists) plus an E11-E14 code [16]. Endometriosis was defined as presence of a diagnostic N80X code (ICD-10) with associated procedures that included fulguration (R4165), ovarian cystectomy (R4421, R4430), adnexectomy (R4331, R4332), and hysterectomy (R4147, R4148, R0141, R0142, R4130, R4149, R4140) within 60 days before or after an initial diagnostic code. ICD-10 codes for abnormal uterine bleeding were N93, N93.8, and N93.9. Progestational agents comprised medroxyprogesterone acetate and megestrol acetate.

All data collection was performed in parallel for both non-atypical and atypical EH. EH type; age at diagnosis of EH; presence of type 2 diabetes, endometriosis, or abnormal uterine bleeding; a diagnostic code for EC; tamoxifen use; types and use of hormone therapy (levonorgestrel-releasing intrauterine system [LNG-IUS] or progestational agents); number of follow-up biopsies after diagnosis of EH; and time from diagnosis of EH to diagnosis of EC were extracted.

## Statistical analyses

SAS® Enterprise Guide® version 6.1 (SAS Institute, Inc., Cary, NC, USA) was used for data mining and analysis. Categorical variables were reported as number and percentage, and continuous variables were reported as mean ± standard deviation (SD). The categorical variables were analyzed using the Chi-square test or Fisher's exact test, whereas the continuous variables were analyzed using the two-tailed independent t test or one-way analysis of variance

(ANOVA) followed by Bonferroni's correction. In addition, the associations of variables with EC in each EH type were analyzed using the Cox Proportional Hazard Regression model with or without adjusting for confounding factors. All variables were used as confounding factors. *P* values <0.05 were considered statistically significant.

## Results

Data from 62,333 women were entered into the database, and the data from 13,744 women met the selection criteria (Fig 1). The data from 11,525 women with non-atypical EH and 2,219 women with atypical EH were extracted for analysis.

### Baseline characteristics of women with endometrial hyperplasia

In total, 48.2% of women with EH were diagnosed between 40–49 years of age. Younger and older ages had lower incidences of EH. Age at diagnosis of EH was not different

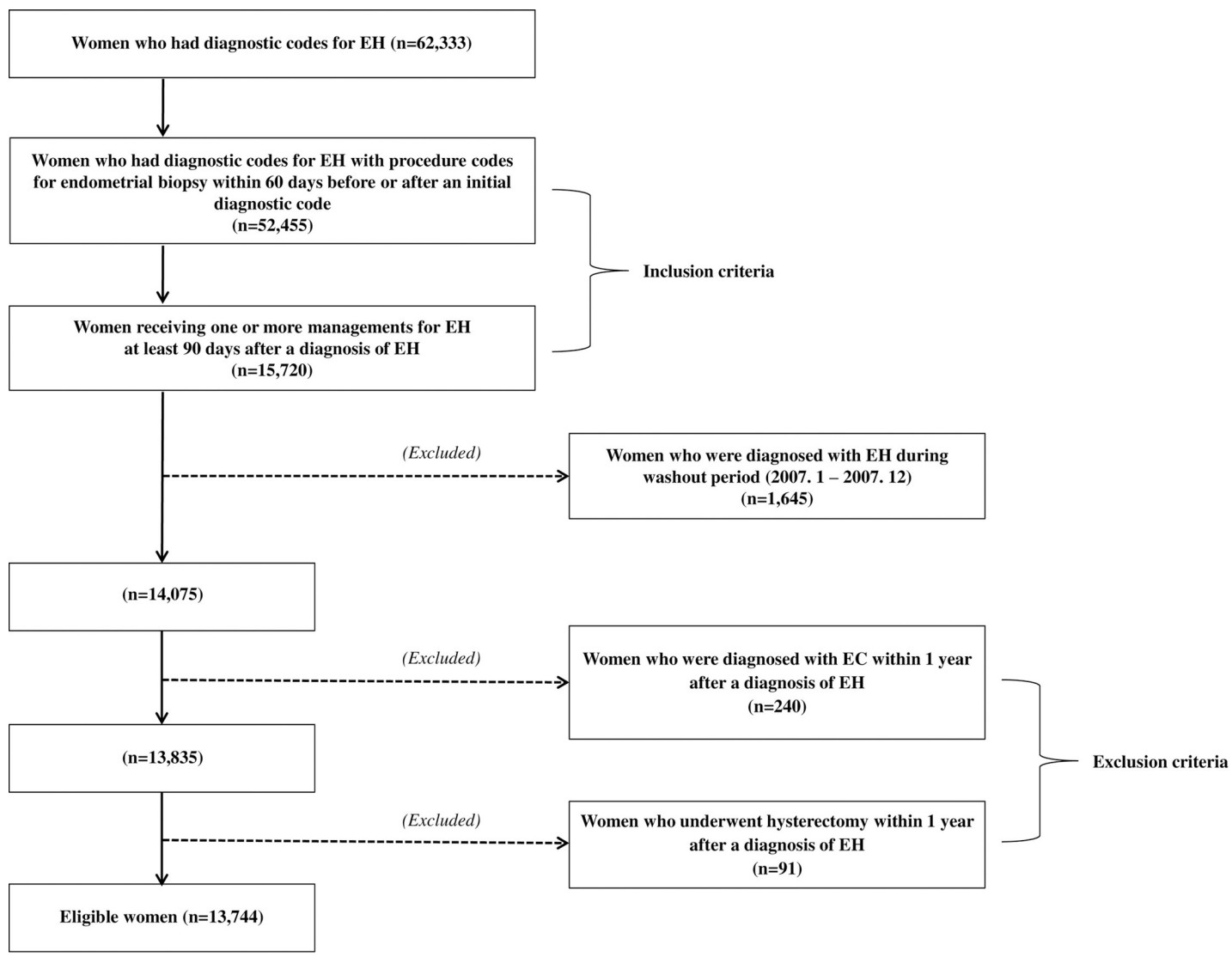

**Fig 1. Flow chart showing selection of eligible patients.**

between women with non-atypical and atypical EH. Type 2 diabetes and endometriosis increased in women with atypical EH compared to those with non-atypical EH. In addition, 30.9% of women with EH did not receive follow-up biopsy after diagnosis of EH, and the frequency of this result was similar in the 2 EH types. Of note, 34.6% of those subjects received only 1 follow-up biopsy, and the frequency was higher in women with non-atypical EH. 3 and ≥4 follow-up biopsies were performed more frequently in women with atypical EH. Abnormal uterine bleeding, tamoxifen use, and use of hormone therapy (LNG-IUS and progestational agents) were not different between women with non-atypical and atypical EH (Table 1). Data for oral contraceptive use were not available.

**Table 1. Baseline characteristics of women with endometrial hyperplasia.**

| | Total | Non-atypical EH | Atypical EH | *P* value |
|---|---|---|---|---|
| | n = 13744 | n = 11525 (83.9%) | n = 2219 (16.1%) | |
| **Age at diagnosis of EH (years), n (%)** | | | | |
| <30 | 1079 (7.9) | 892 (7.8) | 187 (8.4) | 0.369 |
| 30–39 | 2959 (21.5) | 2460 (21.3) | 499 (22.5) | |
| 40–49 | 6628 (48.2) | 5589 (48.5) | 1039 (46.8) | |
| 50–59 | 2394 (17.4) | 2018 (17.5) | 376 (17.0) | |
| ≥60 | 684 (5.0) | 566 (4.9) | 118 (5.3) | |
| **Type 2 diabetes, n (%)** | | | | |
| No | 11257 (81.9) | 9525 (82.7) | 1728 (77.8) | <0.001 |
| Yes | 2487 (18.1) | 1999 (17.3) | 491 (22.2) | |
| **Endometriosis, n (%)** | | | | |
| No | 10465 (76.1) | 8818 (76.5) | 1647 (74.2) | 0.020 |
| Yes | 3279 (23.9) | 2707 (23.5) | 572 (25.8) | |
| **Abnormal uterine bleeding, n (%)** | | | | |
| No | 4859 (35.4) | 4088 (35.5) | 771 (34.8) | 0.513 |
| Yes | 8885 (64.6) | 7437 (64.5) | 1448 (65.2) | |
| **Use of tamoxifen, n (%)** | | | | |
| No | 13607 (99.0) | 11411 (99.0) | 2196 (99.0) | 0.837 |
| Yes | 137 (1.0) | 114 (1.0) | 23 (1.0) | |
| **Hormone therapy, n (%)** | | | | |
| LNG-IUS[a] | | | | |
| No | 12699 (92.4) | 10657 (92.5) | 2042 (92.0) | 0.469 |
| Yes | 1045 (7.6) | 868 (7.5) | 177 (7.0) | |
| Progestational agents | | | | |
| No | 6820 (49.6) | 5727 (49.7) | 1093 (49.3) | 0.707 |
| Yes | 6924 (50.4) | 5798 (50.3) | 1126 (50.7) | |
| **No. of follow-up biopsies after diagnosis of EH, n (%)** | | | | |
| 0 | 4244 (30.9) | 3560 (30.9) | 684 (30.8) | <0.001 |
| 1 | 4751 (34.6) | 4074 (35.3) | 677 (30.5) | |
| 2 | 2466 (17.9) | 2060 (17.9) | 406 (18.3) | |
| 3 | 1167 (8.5) | 942 (8.2) | 225 (10.2) | |
| ≥4 | 1116 (8.1) | 889 (7.7) | 227 (10.2) | |

EH, endometrial hyperplasia

[a]Levonorgestrel-releasing intrauterine system

## Incidence and distribution of endometrial cancer in women with endometrial hyperplasia

Incidence of EC showed a higher tendency in women with atypical EH than in those with non-atypical EH. The C54.1 diagnostic code of endometrial EC was identified in 88.7% of EC cases. Incidence of EC diagnosed with C54.1 was not different between EH types (Table 2). In 1< and ≤ 3 years after diagnosis of EH, EC occurred more frequently in women with non-atypical EH compared to those with atypical EH. EC occurred more frequently in women with atypical EH 3 years after diagnosis of EH than for those with non-atypical EH. However, in cases diagnosed at 5< and ≤ 6 years, EC occurred similarly in both non-atypical and atypical EH. Moreover, contrary to atypical EH, in women with non-atypical EH, EC occurred most frequently at 1< and ≤ 3 years after diagnosis of EH and decreased thereafter (Table 2).

## Risk factor association with endometrial cancer in women with endometrial hyperplasia

Both women with non-atypical and atypical EH showed significant decrease in risk of EC at 40–49 years old compared to women ≤39 years. However, risk of EC was not significantly different between women ≤39 and those ≥50 years old; the incidence of EC increased in women with type 2 diabetes. However, type 2 diabetes was not a risk factor for EC; risk of EC tended to decrease in women with endometriosis when adjusted for other confounding factors (Table 3).

**Table 2. Incidence and distribution of endometrial cancer in women with endometrial hyperplasia.**

| | Total | Non-atypical EH | Atypical EH | P value |
|---|---|---|---|---|
| | n = 13744 | n = 11525 (83.9%) | n = 2219 (16.1%) | |
| **EC, n (%)** | | | | |
|   No | 13294 (96.7) | 11161 (96.8) | 2133 (96.1) | 0.082 |
|   Yes | 450 (3.3) | 364 (3.2) | 86 (3.9) | |
| **EC (C54.1), n (%)** | | | | |
|   No | 13345 (97.1) | 11201 (97.2) | 2144 (96.6) | 0.144 |
|   Yes | 399 (2.9) | 324 (2.8) | 75 (3.4) | |
| **Distribution of EC according to diagnostic code, n (%)** | | | | |
|   C54 | 0 | 0 | 0 | - |
|     C54.0 (Isthmus uteri) | 3 (0.7) | 3 (0.8) | 0 | 0.447 |
|     C54.1 (Endometrium) | 399 (88.7) | 324 (89.0) | 75 (87.2) | 0.144 |
|     C54.2 (Myometrium) | 0 | 0 | 0 | - |
|     C54.3 (Fundus uteri) | 0 | 0 | 0 | - |
|     C54.8 (Overlapping lesion of corpus uteri) | 3 (0.7) | 3 (0.8) | 0 | 0.447 |
|     C54.9 (Corpus uteri, unspecified) | 33 (7.3) | 27 (7.5) | 6 (7.0) | 0.750 |
|   C55 (Malignant neoplasm of uterus, part unspecified) | 12 (2.6) | 7 (1.9) | 5 (5.8) | 0.016 |
| **Distribution of EC according to time from diagnosis of EH to diagnosis of EC (years), n (%)** | | | | |
|   1< and ≤ 2 | 94 (20.9) | 86 (23.6) | 8 (9.3) | 0.038 |
|   2< and ≤ 3 | 91 (20.2) | 77 (21.2) | 14 (16.3) | |
|   3< and ≤ 4 | 72 (16.0) | 53 (14.6) | 19 (22.1) | |
|   4< and ≤ 5 | 67 (14.9) | 51 (14.0) | 16 (18.6) | |
|   5< and ≤ 6 | 43 (9.5) | 35 (9.6) | 8 (9.3) | |
|   6< and ≤ 7 | 25 (5.6) | 18 (4.9) | 7 (8.1) | |
|   7< | 58 (12.9) | 44 (12.1) | 14 (16.3) | |

EH, endometrial hyperplasia; EC, endometrial cancer

**Table 3. Association of risk factors with endometrial cancer occurrence in women with endometrial hyperplasia.**

| | Non-atypical EH n = 11525 (83.9%) | | | | | | | Atypical EH n = 2219 (16.1%) | | | | | | |
|---|---|---|---|---|---|---|---|---|---|---|---|---|---|---|
| | Non-EC n = 11161 (96.8%) | EC n = 364 (3.2%) | P value | Univariate analysis | | Multivariate analysis[b] | | Non-EC n = 2133 (96.1%) | EC n = 86 (3.9%) | P value | Univariate analysis | | Multivariate analysis[b] | |
| | | | | HR (95% CI) | P value | HR (95% CI) | P value | | | | HR (95% CI) | P value | HR (95% CI) | P value |
| **Age at diagnosis of EH (years), n (%)** | | | | | | | | | | | | | | |
| ≤39 | 3202 (28.7) | 150 (41.2) | <0.001 | Reference | | Reference | | 645 (30.2) | 41 (47.7) | | Reference | | Reference | |
| 40–49 | 5463 (49.0) | 126 (34.6) | | 0.511 (0.403–0.648) | <0.001 | 0.557 (0.439–0.708) | <0.001 | 1018 (47.7) | 21 (24.4) | < 0.001 | 0.337 (0.199–0.570) | <0.001 | 0.391 (0.229–0.670) | 0.001 |
| ≥50 | 2496 (22.3) | 88 (24.2) | | 0.807 (0.621–1.050) | 0.111 | 0.954 (0.721–1.261) | 0.739 | 470 (22.0) | 24 (27.9) | | 0.797 (0.481–1.319) | 0.377 | 1.113 (0.630–1.968) | 0.712 |
| **Type 2 diabetes, n (%)** | | | | | | | | | | | | | | |
| No | 9260 (83.0) | 269 (73.9) | <0.001 | Reference | | Reference | | 1669 (78.3) | 59 (68.6) | 0.035 | Reference | | Reference | |
| Yes | 1901 (17.0) | 95 (26.1) | | 1.187 (0.938–1.501) | 0.154 | 1.159 (0.915–1.468) | 0.222 | 464 (21.7) | 27 (31.4) | | 1.243 (0.788–1.962) | 0.350 | 1.155 (0.727–1.835) | 0.543 |
| **Endometriosis, n (%)** | | | | | | | | | | | | | | |
| No | 8542 (76.5) | 276 (75.8) | 0.753 | Reference | | Reference | | 1580 (74.1) | 67 (77.9) | 0.426 | Reference | | Reference | |
| Yes | 2619 (23.5) | 88 (24.2) | | 0.899 (0.707–1.142) | 0.383 | 0.797 (0.624–1.017) | 0.068 | 553 (25.9) | 19 (22.1) | | 0.738 (0.443–1.228) | 0.245 | 0.634 (0.377–1.065) | 0.085 |
| **Abnormal uterine bleeding, n (%)** | | | | | | | | | | | | | | |
| No | 3992 (35.8) | 96 (26.4) | 0.001 | Reference | | Reference | | 741 (34.7) | 30 (34.9) | | Reference | | Reference | |
| Yes | 7169 (64.2) | 268 (73.6) | | 1.339 (1.060–1.691) | 0.014 | 1.123 (0.880–1.433) | 0.350 | 1392 (65.3) | 56 (65.1) | 0.978 | 0.961 (0.617–1.498) | 0.862 | 0.749 (0.470–1.195) | 0.226 |
| **Use of tamoxifen, n (%)** | | | | | | | | | | | | | | |
| No | 11055 (99.1) | 356 (97.8) | 0.018 | Reference | | Reference | | 2111 (99.0) | 85 (98.8) | | Reference | | Reference | |
| Yes | 106 (0.9) | 8 (2.2) | | 1.747 (0.867–3.523) | 0.119 | 1.523 (0.751–3.086) | 0.243 | 22 (1.0) | 1 (1.2) | 0.906 | 1.199 (0.167–8.609) | 0.857 | 1.315 (0.180–0.963) | 0.788 |
| **Hormone therapy, n (%)** | | | | | | | | | | | | | | |
| LNG-IUS[a] | | | | | | | | | | | | | | |
| No | 10330 (92.5) | 327 (89.8) | 0.053 | Reference | | Reference | | 1966 (92.2) | 76 (88.4) | 0.202 | Reference | | Reference | |
| Yes | 831 (7.5) | 37 (10.2) | | 1.463 (1.041–2.055) | 0.028 | 1.265 (0.892–1.795) | 0.187 | 167 (7.8) | 10 (11.6) | | 1.836 (0.949–3.553) | 0.712 | 1.606 (0.806–3.197) | 0.178 |
| Progestational agents | | | | | | | | | | | | | | |
| No | 5545 (49.7) | 182 (50.0) | 0.905 | Reference | | Reference | | 1057 (49.6) | 36 (41.9) | 0.162 | Reference | | Reference | |

(Continued)

**Table 3.** (Continued)

| | Non-atypical EH n = 11525 (83.9%) | | | | | | | Atypical EH n = 2219 (16.1%) | | | | | | |
| | Non-EC n = 11161 (96.8%) | EC n = 364 (3.2%) | P value | Univariate analysis | | Multivariate analysis[b] | | Non-EC n = 2133 (96.1%) | EC n = 86 (3.9%) | P value | Univariate analysis | | Multivariate analysis[b] | |
| | | | | HR (95% CI) | P value | HR (95% CI) | P value | | | | HR (95% CI) | P value | HR (95% CI) | P value |
|---|---|---|---|---|---|---|---|---|---|---|---|---|---|---|
| **Yes** | 5616 (50.3) | 182 (50.0) | | 0.912 (0.742–1.120) | 0.377 | 0.703 (0.565–0.876) | 0.002 | 1076 (50.4) | 50 (58.1) | | 1.376 (0.896–2.112) | 0.144 | 1.022 (0.630–1.658) | 0.929 |
| **No. of follow-up biopsies after diagnosis of EH, n (%)** | | | | | | | | | | | | | | |
| **0** | 3516 (31.5) | 44 (12.1) | <0.001 | Reference | | Reference | | 672 (31.5) | 12 (14.0) | <0.001 | Reference | | Reference | |
| **1** | 3974 (35.6) | 100 (27.5) | | 1.740 (1.220–2.480) | 0.002 | 1.835 (1.282–2.629) | 0.001 | 658 (30.8) | 19 (22.1) | | 1.641 (0.796–3.380) | 0.180 | 1.825 (0.871–3.824) | 0.111 |
| **≥2** | 3671 (32.9) | 220 (60.4) | | 3.303 (2.389–4.567) | <0.001 | 3.644 (2.585–5.317) | <0.001 | 803 (37.7) | 55 (63.9) | | 3.337 (1.787–6.231) | 0.001 | 3.827 (1.924–7.612) | 0.001 |

EH, endometrial hyperplasia; EC, endometrial cancer; HR, hazard ratio; CI, confidence interval

[a]Levonorgestrel-releasing intrauterine system.

[b]The data were adjusted for all risk factors (age at diagnosis of EH, type 2 diabetes, endometriosis, abnormal uterine bleeding, use of tamoxifen, LNG-IUS, progestational agents, and number of follow-up biopsies after diagnosis of EH).

Women with non-atypical EH showed increase in EC in women with abnormal uterine bleeding and LNG-IUS. However, there were no risk factors for EC when adjusted for other confounding factors, and the incidence of EC increased in women using tamoxifen. However, tamoxifen use was not a risk factor for EC; progestational agents were associated with significant decrease of EC risk when adjusted for other confounding factors, and the risk of EC significantly increased according to number of follow-up biopsies after diagnosis of EH (Table 3).

The following findings were identified for women with atypical EH: abnormal uterine bleeding, tamoxifen use, and use of hormone therapy (LNG-IUS and progestational agents) were not associated with EC and were not risk factors for EC, and the risk of EC significantly increased when ≥2 follow-up biopsies were performed after diagnosis of EH (Table 3).

## Incidence of and progression time to endometrial cancer according to age at diagnosis of endometrial hyperplasia

The cumulative incidence of EC was associated with a higher tendency in women with atypical EH than in those with non-atypical EH (P = 0.082). The incidence density of EC was not significantly different between women with non-atypical and atypical EH (P = 0.913). Cumulative incidence and incidence density of EC according to age at diagnosis of EH were associated with the following findings: in women with non-atypical EH, the lowest incidence occurred in women 40–49 years of age and the highest in women ≥70 years old; in women with atypical EH, the lowest incidence was for women 40–49 years of age and the highest was in women 30–39 or ≥70 years of age (Table 4).

Time to progression to EC in women with non-atypical EH decreased in women ≥50 years old compared to other ages (≤39 and 40–49 years old). Time to progression to EC in women with atypical EH was not different among ages (Table 4).

**Table 4. Incidence of and progression time to endometrial cancer according to age at diagnosis of endometrial hyperplasia.**

| | Non-atypical EH | | | Atypical EH | | |
|---|---|---|---|---|---|---|
| | Total n = 11525 | EC n = 364 | Incidence (95% CI) | Total n = 2219 | EC n = 86 | Incidence (95% CI) |
| **Cumulative incidence (per 1000 persons)** | | | | | | |
| **Total** | 11525 | 364 | 31.6 (25.8–34.9) | 2219 | 86 | 38.8 (31.3–47.4) |
| **Age at diagnosis of EH (years)** | | | | | | |
| **<30** | 892 | 40 | 44.8 (3.8–62.0) | 187 | 8 | 42.8 (20.1–79.6) |
| **30–39** | 2460 | 110 | 44.7 (37.1–53.4) | 499 | 33 | 66.1 (46.7–90.6) |
| **40–49** | 5589 | 126 | 22.5 (18.9–26.7) | 1039 | 21 | 20.2 (12.9–30.2) |
| **50–59** | 2018 | 66 | 32.7 (25.6–41.2) | 376 | 18 | 47.9 (29.5–73.2) |
| **60–69** | 406 | 14 | 34.5 (19.8–55.3) | 87 | 4 | 46.0 (14.8–107.2) |
| **≥70** | 106 | 8 | 75.5 (36.7–138.3) | 31 | 2 | 64.5 (11.0–197.2) |
| **Incidence density (per 1000 person years)** | | | | | | |
| **Total** | 56,883.4 | 364 | 6.4 (5.5–7.1) | 13,264.3 | 86 | 6.5 (5.2–8.0) |
| **Age at diagnosis of EH (years)** | | | | | | |
| **<30** | 4307.8 | 40 | 9.3 (6.7–12.5) | 1090.2 | 8 | 7.3 (3.4–13.9) |
| **30–39** | 12513.3 | 110 | 8.8 (7.3–10.5) | 2971.0 | 33 | 11.1 (7.8–15.4) |
| **40–49** | 27663.5 | 126 | 4.6 (3.8–5.4) | 6204.5 | 21 | 3.4 (2.2–5.1) |
| **50–59** | 9733.5 | 66 | 6.8 (5.3–8.6) | 2298.4 | 18 | 7.8 (4.8–12.1) |
| **60–69** | 1892.2 | 14 | 7.4 (4.2–12.1) | 518.0 | 4 | 7.7 (2.5–18.5) |
| **≥70** | 689.0 | 8 | 11.6 (5.4–21.9) | 182.3 | 2 | 11.0 (1.8–35.8) |
| **Time from diagnosis of EH to diagnosis of EC (years), mean ±SD** | | | | | | |
| **Age at diagnosis of EH (years)** | | | | | | |
| **≤39** | 3.95 ± 2.18 | | P = 0.004 | 4.38 ± 2.31 | | P = 0.576 |
| **40–49** | 4.11 ± 2.32 | | | 4.26 ± 1.94 | | |
| **≥50** | 3.22 ± 1.88 | | | 4.89 ± 2.28 | | |

EH, endometrial hyperplasia; EC, endometrial cancer

## Discussion

### Summary

In this claims data analysis of 11,525 women with EH, women ≤39 and ≥50 years with EH were at high risk for progression to EC. Detection of EC increased in proportion to number of follow-up biopsies performed after diagnosis of EH. Time to progression to EC was shorter in women ≥50 years old with non-atypical EH and similar among ages in women with atypical EH. Progestational agents were a protective factor for progression to EC only in women with non-atypical EH. Endometriosis had a potentially protective role for EC. However, type 2 diabetes, abnormal uterine bleeding, tamoxifen use, and LNG-IUS were not risk factors for progression to EC.

### Interpretation

**Incidence of endometrial cancer.** Retrospective studies have reported that atypical EH is associated with a higher risk for concurrent EC or progression to EC compared to non-atypical EH [2, 3, 8–11]. However, in this study, the incidence of EC showed a higher tendency only in women with atypical EH. This difference might be attributed to use of strict criteria for eligible women, followed by potentially excessive exclusion of data because of the use of claims data and because medical records cannot be reviewed.

A nested case-control study reported that cumulative risks of progression to EC increased with time in both EH types and were higher in women with atypical EH than in those with non-atypical EH [8]. However, in this study, at 1< and ≤ 3 years after diagnosis of non-atypical EH, EC occurred most frequently and more frequently than in women with atypical EH. These findings suggest that EC diagnosed in the first 3 years after diagnosis of non-atypical EH may be cancer already present at the first diagnostic procedures such as endometrial sampling, dilatation and curettage, and hysteroscopic guided endometrial biopsy, which are usually used to diagnose EH and EC in our country.

**Age at diagnosis of endometrial hyperplasia: A risk factor.** Retrospective studies have reported that women ≥50–53 years old with EH were independent predictors of concurrent EC [9, 11]. In a retrospective case–control study, concurrent EC increased in women 40–59 and ≥60 years of age with EH compared to such women <40 years old [10]. In this study, risk of progression to EC in women with EH decreased in the 40–49 years group, suggesting that women ≤39 and ≥50 years old with EH were at higher risk for progression to EC. Moreover, cumulative incidence and incidence density of EC in women with EH were lowest in women 40–49 years old and highest in women ≥70 years old, and the highest level was observed in women 30–39 years old with atypical EH. Our findings are similar to those of a previous study in Korea that reported that the annual incidence rate of EC has been increasing (annual percent changes [APC], 6.9% during 1999–2010). In that study, women <30 years old had the highest APC (11.2%), women ≥80 years old had the second highest APC (9.5%), and women 40–49 and 70–79 years old had the lowest APC (5.3% and 5.6%, respectively) [6]. In our study, it is possible that women 40–49 years old had relatively mild EH (i.e., simple > complex) because most (48.2%) were diagnosed at an age of 40–49 years, suggesting a tendency to actively undergo examination. It is also possible that younger or older women had relatively severe EH (i.e., simple < complex) because they tended to visit the hospital when they experienced symptoms.

In addition, in this study, progression to EC occurred more quickly in women ≥50 years old with non-atypical EH. This finding supports the hypothesis that women ≥50 years old with non-atypical EH have a high risk for progression to EC.

**Follow-up biopsy after diagnosis of endometrial hyperplasia: A protective factor.** This study showed that detection of EC in women with EH increased in proportion to an increase in number of follow-up biopsies. However, 30.9% of women with EH did not undergo follow-up biopsy after diagnosis of EH. Moreover, reflecting a more serious concern for EC in women with atypical EH, women with atypical EH received more follow-up biopsies than women with non-atypical EH. Therefore, if women with EH receive repeated follow-up biopsies, it may increase the detection rate of progression to EC.

## Progestational agents: A protective factor

Oral progestational agents have been a popular therapeutic choice in women with non-atypical EH and have been used as conservative treatment in women with atypical EH [17, 18]. In meta-analyses of randomized trials, LNG-IUS achieved higher regression rates and lower hysterectomy rates than progestational agents in women with non-atypical EH and both EH types [17, 19]. In a small retrospective study (n = 48), LNG-IUS achieved high regression rates in women with atypical complex hyperplasia or EC [20]. However, in this study, LNG-IUS was not associated with progression to EC in women with EH; in addition, progestational agents had a protective role for progression to EC only in women with non-atypical EH and did not impact women with atypical EH. These findings might be attributed to the severity of EH among women that used LNG-IUS. In this study, in women with non-atypical EH, LNG-IUS

had been used in 7.5% of those diagnosed as non-EC and 10.2% of those diagnosed as EC; in women with atypical EH, LNG-IUS had been used in 7.8% of those diagnosed as non-EC and 11.6% of those diagnosed as EC.

**Endometriosis and other factors.** Endometriosis is a common gynecologic disease, affecting 5–15% of premenopausal women and 2.2–5% of postmenopausal women [21, 22]. EC has been reported to occur in 0.17–6.7% of women with endometriosis [23–25]. Although it is not clear whether endometriosis is a risk factor for EC, some large-scale studies have reported a significantly increased risk for endometrial cancer in women with a diagnosis of endometriosis [23–25]. In this study, women with atypical EH experienced endometriosis more frequently than women with non-atypical EH. Additionally, risk of progression to EC in women with EH tended to decrease in women with endometriosis. We presume that the use of progestational agents for treatment of endometriosis might contribute to our finding suggesting that progestational agents have a protective role against EC in women with EH. This study is the first to evaluate the relationship between EH and endometriosis. Therefore, results of this study should be evaluated further in subsequent studies.

Diabetes mellitus and/or high BMI ($\geq$ 35 or $\geq 27$ kg/m$^2$) have been reported as predictive factors of concurrent EC in women with EH [10, 11]. In this study, women with atypical EH had type 2 diabetes more frequently than women with non-atypical EH. However, type 2 diabetes was not a risk factor for progression to EH.

Abnormal uterine bleeding is the most common symptom of EH. Some studies have reported that, based on endometrial biopsy, about 70%, 15%, and 15% of women with abnormal uterine bleeding are diagnosed with benign findings, EH, and EC, respectively [26]. In this study, abnormal uterine bleeding occurred similarly between women with non-atypical and atypical EH and was not a risk factor for progression to EH.

The incidence (1.3–20%) of EH has been noted to increase in postmenopausal women with breast cancer treated with tamoxifen compared to the incidence (0–10%) in those who did not receive tamoxifen [27]. Various studies have reported that tamoxifen use induces a 1.3–7.5-fold increase in the relative risk of endometrial cancer [27]. However, a retrospective study reported that, in 333 peri/postmenopausal women with breast cancer, incidences of EH and EC were lower in women treated with tamoxifen compared to those who did not receive tamoxifen (EH: 3% vs. 11.1%; EC: 3.8% vs. 11.1%) [28]. Moreover, in a recent multicenter retrospective cohort study (n = 1129), there were no differences in incidences of EH and EC among women with breast cancer who received tamoxifen, aromatase inhibitors, and no treatment [29]. In this study, tamoxifen use was not different between women with non-atypical and atypical EH and was not a risk factor for progression to EH, increasing controversy about the role of tamoxifen as an inducer of EC.

**Patient selection.** Many women are assigned an EH diagnostic code before they undergo endometrial biopsy. Some of them continue to have diagnostic codes of EH during follow-up, although the pathology does not show EH. Based on a review of medical records in Inha University Hospital, women with continuous EH diagnostic codes without EH pathology were followed for a mean of 90 days after the initial EH diagnostic code. Therefore, we only included women in this study that had EH diagnostic codes and were followed up for at least 90 days after initial EH diagnostic code, to adjust for these diagnostic inconsistencies and to generate a more appropriate study population.

Moreover, we excluded women who were diagnosed with EC within 1 year after diagnosis of EH to exclude women with concurrent EC. We also excluded women who underwent hysterectomy within 1 year after diagnosis of EH because this study intended to evaluate risk and risk factors of progression to EC in women with EH.

## Strengths and limitations

The significance of this study was the ability to verify risk factors that are related to progression to EC in women with non-atypical and atypical EH. To our knowledge, this is the first report that identifies clinical risk factors that predict progression to EC in women with EH.

There were some limitations to this study based on use of claim data. First, because diseases in this study were indirectly defined based on diagnosis, procedure, and drug codes, some cases may have been incorrectly diagnosed by erroneous coding. However, women with EH and endometriosis were only selected when pathologic examinations had been performed, and EC was only selected when women had pathologic examinations or at least 2 reliable EC codes. Moreover, the type 2 diabetes indication followed the definition of a previous published study [15]. Therefore, a few incorrectly diagnosed cases might have been included in this study. Second, the population with EH could not be accurately selected because medical records could not be reviewed. However, we selected those cases based on review of medical records in our hospital to exclude women with EH diagnostic codes that were not supported by pathology. Therefore, it is likely that only a limited number of incorrect cases was analyzed in this study. Third, although we intended to exclude women with EH with concurrent EC, a small percentage of women with non-atypical EH and concurrent EC might have been included in our cohort. Finally, this study could not be considered for central pathologic review because the HIRA dataset uses anonymous identification codes.

## Conclusions

Based on claim data analysis, we demonstrated that, regardless of EH type, women ≤39 and ≥50 years of age with EH had high risk for progression to EC. Moreover, this study indicates the importance of repeated follow-up biopsy after diagnosis of EH regardless of EH type. Finally, different influences of progestational agents for progression to EC depending on EH type may support current trends for non-surgical management in women with non-atypical EH and surgical management in women with atypical EH.

## Author Contributions

**Conceptualization:** Banghyun Lee, Kidong Kim.

**Data curation:** Jin Young Jeong, Sung Ook Hwang, Banghyun Lee.

**Formal analysis:** Jin Young Jeong.

**Funding acquisition:** Banghyun Lee.

**Investigation:** Jin Young Jeong, Sung Ook Hwang, Banghyun Lee, Kidong Kim, Yong Beom Kim, Sung Hye Park, Hwa Yeon Choi.

**Methodology:** Banghyun Lee, Kidong Kim.

**Project administration:** Banghyun Lee.

**Resources:** Banghyun Lee.

**Supervision:** Banghyun Lee.

**Writing – original draft:** Jin Young Jeong, Sung Ook Hwang, Banghyun Lee.

**Writing – review & editing:** Jin Young Jeong, Sung Ook Hwang, Banghyun Lee, Kidong Kim, Yong Beom Kim, Sung Hye Park, Hwa Yeon Choi.

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
