## [Decision Letter · Decision Letter 0]

21 Oct 2020

PONE-D-20-17525

Risk factors of progression to endometrial cancer in women with endometrial hyperplasia: A retrospective cohort study

PLOS ONE

Dear Dr. Lee,

Thank you for submitting your manuscript to PLOS ONE. After careful consideration, we feel that it has merit but does not fully meet PLOS ONE’s publication criteria as it currently stands. Therefore, we invite you to submit a revised version of the manuscript that addresses the points raised during the review process.

I think the manuscript is interesting. Please address all reviewers’ comments in order to improve its quality.

We look forward to receiving your revised manuscript.

Kind regards,

Diego Raimondo

Academic Editor

PLOS ONE

Journal Requirements:

2.We note that you have indicated that data from this study are available upon request. PLOS only allows data to be available upon request if there are legal or ethical restrictions on sharing data publicly. For information on unacceptable data access restrictions, please see http://journals.plos.org/plosone/s/data-availability#loc-unacceptable-data-access-restrictions.

Reviewers' comments:

Reviewer's Responses to Questions

**Comments to the Author**

1. Is the manuscript technically sound, and do the data support the conclusions?

Reviewer #1: Yes

Reviewer #2: Yes

2. Has the statistical analysis been performed appropriately and rigorously? 

Reviewer #1: Yes

Reviewer #2: Yes

3. Have the authors made all data underlying the findings in their manuscript fully available?

Reviewer #1: No

Reviewer #2: Yes

4. Is the manuscript presented in an intelligible fashion and written in standard English?

Reviewer #1: Yes

Reviewer #2: Yes

5. Review Comments to the Author

Reviewer #1: Dear Authors,

thank you for submitting your Manuscript to “Plos One” journal.

I think the manuscript is interesting. The statistical analysis is well conducted and even if this work does not add great novelties to the knowledge of both EH and EC, the wide clinical sample helps to underline some risk factor of progression from endometrial hyperplasia to endometrial cancer.

Nevertheless, I have the follow comments:

1) According to PLOS editorial policy, all data underlying the findings described in your manuscript should be fully available without restriction. You affirm that by HIRA policy, data cannot be shared. You should have extracted the data regarding the included patients and let them available in an anonymous way, at least those regarding the investigated risk factors.

2) Some minor English mistakes should be corrected.

3) Please update bibliography, some citations are not up-to-date. Regarding endometriosis in post-menopausal women see:

Hormonal Replacement Therapy in Menopausal Women with History of Endometriosis: A Review of Literature. Zanello M et al. Medicina (Kaunas). 2019 Aug 14;55(8):477. doi: 10.3390/medicina55080477.

Regarding conservative treatment of EH, it has been proposed also for EC: see Conservative hysteroscopic treatment of stage I well differentiated endometrial cancer in patients with high surgical risk: a pilot study. Casadio P et al. J Gynecol Oncol. 2019 Jul;30(4):e62. doi: 10.3802/jgo.2019.30.e62. Epub 2019 Apr 22

4) In the “Conclusion” section you affirm that “Moreover, this study indicates the necessity of repeated follow-up biopsy after diagnosis of EH regardless of EH type.” (lines 341-343). This study does not indicate the necessity of biopsies; it just underlines the link between repeated biopsies and EC detection, suggesting the opportunity of repeated biopsies.

Reviewer #2: The topic of this Manuscript falls within the aims of PLOS ONE. The article provides a valuable and methodologically correct data analysis, the conclusion is consistent with data discussion and with available evidence throughout the text. Presentation of the Manuscript conforms Journal’s guideline for Authors. The topic of this study is actual, so it may be of interest for the readers.

In general, the Manuscript may benefit from some minor revisions, as suggested below:

- I don’t think that non-atypical hyperplasia evolves towards endometrial cancer more frequently in the first 3 years after diagnosis. How do you explain your data? Maybe your data could be justified by the fact that these cancers, diagnosed in the first 3 years, are actually misdiagnosed cancers already present at the first hysteroscopy. Please, indicate your opinion and specify how you in Korea usually perform the diagnosis of EH and EC (guided hysteroscopic biopsy or blind biopsy).

- How do you explain the protective role of endometriosis? Maybe the use of progestin agents in this pathology contributes to reduce the incidence of endometrial cancer in this population. Please, discuss these data and specify your opinion.

- Talking about the role of tamoxifen, in my opinion these recent studies that don’t show an increased risk of EC in women treated with tamoxifen, have to be cited. PMID: 23599784; PMID: 31425735.

- In general the manuscript could benefit of these topic-related citations: PMID: 29382392, PMID: 32226771.

6. PLOS authors have the option to publish the peer review history of their article (what does this mean?). If published, this will include your full peer review and any attached files.

Reviewer #1: No

Reviewer #2: No

---

## [Author Response · Author response to Decision Letter 0]

2 Nov 2020

Reviewer #1: Dear Authors,

thank you for submitting your Manuscript to “Plos One” journal.

I think the manuscript is interesting. The statistical analysis is well conducted and even if this work does not add great novelties to the knowledge of both EH and EC, the wide clinical sample helps to underline some risk factor of progression from endometrial hyperplasia to endometrial cancer.

Nevertheless, I have the follow comments:

1) According to PLOS editorial policy, all data underlying the findings described in your manuscript should be fully available without restriction. You affirm that by HIRA policy, data cannot be shared. You should have extracted the data regarding the included patients and let them available in an anonymous way, at least those regarding the investigated risk factors.

The Health Insurance Review and Assessment Service (HIRA) dataset uses anonymous identification codes to protect patients' personal information. Therefore, if researcher may extract data of patients, data can be available in an anonymous way. However, the HIRA limits use of dataset as follows: 1. Researcher may use the HIRA dataset during specific period (paid by researcher) only using computer with specific Internet Protocol (IP) address permitted by the HIRA; 2. Researcher may have results of analysis and may not have data which are used to analyze in study. The system of the HIRA blocks researcher to take out data out of computer with specific IP address permitted by the HIRA. Because of this reason, we cannot provide data regarding the risk factors investigated in our study.

We sincerely hope that the reviewer and editor would understand this difficult situation.

Moreover, we can find that a lot of articles using the HIRA dataset have been published in PLOS ONE.

2) Some minor English mistakes should be corrected.

Manuscript was rechecked by experts who are skilled authors of English language papers (www.eworldediting.com‎).

3) Please update bibliography, some citations are not up-to-date. Regarding endometriosis in post-menopausal women see:

Hormonal Replacement Therapy in Menopausal Women with History of Endometriosis: A Review of Literature. Zanello M et al. Medicina (Kaunas). 2019 Aug 14;55(8):477. doi: 10.3390/medicina55080477.

Regarding conservative treatment of EH, it has been proposed also for EC: see Conservative hysteroscopic treatment of stage I well differentiated endometrial cancer in patients with high surgical risk: a pilot study. Casadio P et al. J Gynecol Oncol. 2019 Jul;30(4):e62. doi: 10.3802/jgo.2019.30.e62. Epub 2019 Apr 22

We appreciate for your good comment.

Hormonal Replacement Therapy in Menopausal Women with History of Endometriosis: The sentence was corrected as follows: Endometriosis is a common gynecologic disease, affecting 5–15% of premenopausal women and 2.2–5% of postmenopausal women [21, 22].

: ‘affecting 5–15% of premenopausal women’ in our manuscript include ‘affects 10–15% of women of reproductive age’ provided in recommended article. Therefore, it was not changed.

Regarding conservative treatment of EH: The article recommended by reviewer is a prospective pilot study reporting effectiveness and safety of endo-myometrial hysteroscopic resection and the placement of LNG-IUD in women diagnosed with stage IA, grade 1 endometrioid EC. That study reported that in nine women diagnosed with stage IA, grade 1 endometrioid EC which was contraindicated or refused standard treatment with external beam radiation therapy with or without brachytherapy, endo-myometrial hysteroscopic resection of the whole uterine cavity and the placement of LNG-IUD for 5 years showed no recurrence. However, this interesting article does not show specific relationship with our study evaluating risk factors of progression to EC in women with EH including role of progestational agents. Therefore, it is difficult to discuss that article in our study. We sincerely hope that the reviewer would understand this point.

4) In the “Conclusion” section you affirm that “Moreover, this study indicates the necessity of repeated follow-up biopsy after diagnosis of EH regardless of EH type.” (lines 341-343). This study does not indicate the necessity of biopsies; it just underlines the link between repeated biopsies and EC detection, suggesting the opportunity of repeated biopsies.

We appreciate for your good comment.

The sentence was corrected as follows: Moreover, this study indicates the importance of repeated follow-up biopsy after diagnosis of EH regardless of EH type.

Reviewer #2: The topic of this Manuscript falls within the aims of PLOS ONE. The article provides a valuable and methodologically correct data analysis, the conclusion is consistent with data discussion and with available evidence throughout the text. Presentation of the Manuscript conforms Journal’s guideline for Authors. The topic of this study is actual, so it may be of interest for the readers.

In general, the Manuscript may benefit from some minor revisions, as suggested below:

- I don’t think that non-atypical hyperplasia evolves towards endometrial cancer more frequently in the first 3 years after diagnosis. How do you explain your data? Maybe your data could be justified by the fact that these cancers, diagnosed in the first 3 years, are actually misdiagnosed cancers already present at the first hysteroscopy. Please, indicate your opinion and specify how you in Korea usually perform the diagnosis of EH and EC (guided hysteroscopic biopsy or blind biopsy).

We appreciate for your helpful comment.

Following sentence was inserted into discussion: …… These findings suggest that EC diagnosed in the first 3 years after diagnosis of non-atypical EH may be misdiagnosed as cancer already present at the first diagnostic procedures such as endometrial sampling, dilatation and curettage, and hysteroscopic guided endometrial biopsy, which are usually used to diagnose EH and EC in our country. 

- How do you explain the protective role of endometriosis? Maybe the use of progestin agents in this pathology contributes to reduce the incidence of endometrial cancer in this population. Please, discuss these data and specify your opinion.

We appreciate for your good comment.

Following sentence was inserted into discussion: Additionally, risk of progression to EC in women with EH tended to decrease in women with endometriosis. We presume that the use of progestational agents for treatment of endometriosis might contribute to our finding suggesting that progestational agents have a protective role against EC in women with EH. This study is the first to evaluate the relationship between EH and endometriosis. Therefore, results of this study should be evaluated further in subsequent studies. 

- Talking about the role of tamoxifen, in my opinion these recent studies that don’t show an increased risk of EC in women treated with tamoxifen, have to be cited. PMID: 23599784; PMID: 31425735.

We appreciate for your good comment.

Following sentences were inserted into discussion: ……….. However, a retrospective study reported that, in 333 peri/postmenopausal women with breast cancer, incidences of EH and EC were lower in women treated with tamoxifen compared to those who did not receive tamoxifen (EH: 3% vs. 11.1%; EC: 3.8% vs. 11.1%) [28]. Moreover, in a recent multicenter retrospective cohort study (n=1129), there were no differences in incidences of EH and EC among women with breast cancer who received tamoxifen, aromatase inhibitors, and no treatment [29]. In this study, tamoxifen use was not different between women with non-atypical and atypical EH and was not a risk factor for progression to EH, increasing controversy about the role of tamoxifen as an inducer of EC.

- In general the manuscript could benefit of these topic-related citations: PMID: 29382392, PMID: 32226771.

Following sentences were inserted into introduction: ……………. Some studies have reported that serum DNA integrity index and molecular markers and immune cells related to the immune escape mechanisms may play roles in EC [12, 13]. Evaluating these roles in EH might be of benefit to predicting progression to EC in women with EH.

---

## [Editor Report · Decision Letter 1]

16 Nov 2020

Risk factors of progression to endometrial cancer in women with endometrial hyperplasia: A retrospective cohort study

PONE-D-20-17525R1

Dear Dr. Banghyun Lee

We’re pleased to inform you that your manuscript has been judged scientifically suitable for publication and will be formally accepted for publication once it meets all outstanding technical requirements.

Kind regards,

Diego Raimondo

Academic Editor

PLOS ONE

Additional Editor Comments (optional):

I congratulate with the authors for the paper and the adequate comments.

Reviewers' comments:

none

---

## [Editor Report · Acceptance letter]

18 Nov 2020

PONE-D-20-17525R1 

Risk factors of progression to endometrial cancer in women with endometrial hyperplasia: A retrospective cohort study 

Dear Dr. Lee:

I'm pleased to inform you that your manuscript has been deemed suitable for publication in PLOS ONE. Congratulations! Your manuscript is now with our production department. 

Kind regards, 

on behalf of

Dr. Diego Raimondo 

Academic Editor

PLOS ONE